# Antibiofilm Activity of Omega-3 Fatty Acids and Its Influence on the Expression of Biofilm Formation Genes on *Staphylococcus aureus*

**DOI:** 10.3390/antibiotics11070932

**Published:** 2022-07-11

**Authors:** Christopher Spiegel, Stephan Josef Maria Steixner, Débora C. Coraça-Huber

**Affiliations:** Research Laboratory for Biofilms and Implant Associated Infections (BIOFILM LAB), Experimental Orthopaedics, University Hospital for Orthopaedics and Traumatology, Medical University of Innsbruck, Peter-Mayr-Strasse 4b, Room 204, 6020 Innsbruck, Austria; stephan.steixner@i-med.ac.at (S.J.M.S.); debora.coraca-huber@i-med.ac.at (D.C.C.-H.)

**Keywords:** biofilm, biofilm genes, icaADBC, stress responses, SarA, SigB, MRSA, *Staphylococcus aureus*, omega-3 fatty acids, docosahexaenoic acid, antibiofilm, antimicrobial

## Abstract

**Background:** Currently, 1–2% of all prosthetic joint surgeries are followed by an infection. These infections cause approximately 4% of deaths in the first year after surgery, while the 5-year mortality rate is up to 21%. Prosthetic joint infections are mainly caused by *Staphylococcus aureus* or *Staphylococcus epidermis* strains. Both species share the capability of biofilm formation and methicillin resistance. The formation of biofilm helps bacterial cells to withstand critical environmental conditions. Due to their tolerance against antibacterial substances, biofilms are a significant problem in modern medicine. Alternatives for the use of methicillin as a therapeutic are not yet widespread. The use of omega-3 fatty acids, such as docosahexaenoic acid, may help against prosthetic joint infections and lower mortality rates. The aim of this study is to evaluate if docosahexaenoic acid offers a safe anti-biofilm activity against *Staphylococcus aureus* and MRSA without enhancing icaADBC-dependent biofilm formation or additional stress responses, therefore enhancing antibiotic tolerance and resistance. **Methods:** In this study, we examined the gene expression of biofilm-associated genes and regulators. We performed RT-qPCR after RNA extraction of *Staphylococcus aureus* ATCC 29213 and one clinical MRSA strain. We compared gene expression of icaADBC, SarA, SigB, and agrAC under the influence of 1.25 mg /L and 0.625 mg/L of docosahexaenoic acid to their controls. **Results:** We found a higher expression of regulatory genes such as SarA, SigB, agrA, and agrC at 1.25 mg/L of docosahexaenoic acid in ATCC 29213 and a lower increase in gene expression levels in clinical MRSA isolates. icaADBC was not affected in both strains at both concentration levels by docosahexaenoic acid. **Conclusions:** Docosahexaenoic acid does not enhance icaADBC-dependent biofilm formation while still reducing bacterial CFU in biofilms. Docosahexaenoic acid can be considered an option as a therapeutic substance against biofilm formation and may be a good alternative in reducing the risk of MRSA formation.

## 1. Introduction

Bone and implant-related infections are still a big challenge in the field of orthopedic surgery. Approximately 1–2% of all prosthetic joint surgeries are followed by infection [1]. These infections cause approximately 4% of deaths in the first year after surgery, while the 5-year mortality rate is up to 21% [2]. More than 20–50% of open bone fractures are followed by bone infections, and 25% of all clinical infections are related to device-associated infections [3]. Medical device-related infections are mainly caused by *coagulase-negative Staphylococci* (CNS), *Staphylococcus aureus* strains, and methicillin-resistant strain variants of *Staphylococcus aureus* (MRSA), as well as CNS [4]. These bacteria are related to early implant infections (2–8 weeks) as well as to late infections (3–36 months). These microorganisms have the ability to form biofilms, one of the reasons why implant-associated infections are challenging to treat.

*Staphylococci* commonly form biofilm and can develop methicillin resistance. Mechanisms like bacterial cell linkage and the excretion of eDNA stabilize the staphylococcal biofilm [5]. The formation of biofilm helps bacterial cells to withstand critical environmental conditions, e.g., changes in pH, changes in osmotic pressures, and rise in antibiotic substance concentrations. During cell stress, gene expression of the gene regulator sigma B is upregulated [6].

The gene sigma B is linked to the transcriptional regulator staphylococcal regulator A (SarA) and negatively regulates the expression of the accessory gene regulator (Agr). SarA regulates Agr expression and the locus icaADBC [7]. IcaADBC is responsible for the production of polysaccharide intercellular adhesins (PIA). Agr controls the production of phenol soluble modulins (PSMs), extracellular proteases, and toxins [8]. It is also responsible for the quorum sensing mechanisms in biofilm and, therefore, involved in the control of biofilm maturation and bacterial cell density in biofilm [9]. Alternative biofilm-related proteins like fibronectin-binding protein adhesins (FnBPs) and biofilm-associated proteins (Bap) are also regulated by SarA [10]. One of the most important protein groups associated with biofilm formation are PIA and their regulators, icaADBC. PIA connects adjacent bacterial cells together to form biofilms. The expression of PIA is suggested to be a biomarker of biofilm formation [11]. In this study, we focused on the expression of sigma factor, SarA, Agr, and icaADBC.

Due to their tolerance against antibacterial substances, biofilms are a significant problem in modern medicine. The common use of antibiotics in general, but especially methicillin, is leading to high rates of Methicillin-Resistant *Staphylococcus aureus* strains (MRSA). Around 30% of all *Staphylococcus aureus* strains isolated from patients undergoing implant-related infection treatment have a methicillin resistance profile [12]. This high incidence of antibiotic-resistant genes demands the development of alternative antibiotics. In the last couple of years, the use of omega-3 fatty acids as an alternative for the treatment of implant-associated infections has shown promising results [13]. Coraca–Huber et al. showed antibacterial and anti-biofilm formation capabilities of docosahexaenoic acid (DHA) and eicosapentaenoic acid (EPA) against *Staphylococcus aureus* and CNS [14]. Omega-3 polyunsaturated fatty acids (PUFAs) consist of α-Linolenic acid (ALA), docosahexaenoic acid (DHA), and eicosapentaenoic acid (EPA). ALA can be found in plants, whereas DHA and EPA are commonly found in fish or algae. Omega-3 PUFAs are essential fatty acids for humans and are important for cell membrane formation and functionality in mammalian cells [15]. The aim of this study is to evaluate if DHA offers a safe anti-biofilm activity against *Staphylococcus aureus* and MRSA without enhancing icaADBC-dependent biofilm formation or additional stress responses, therefore enhancing the spread of antibiotic tolerance and resistance.

## 2. Materials and Methods

### 2.1. Substances

In this study, we used the PUFA docosahexaenoic acid (DHA; Cayman Chemical Company, Ann Arbor, MI, USA). We used sub-minimal inhibitory concentrations (MIC) of 1.25 mg/L and 0.625 mg/L DHA following the previously performed studies with an MIC of *Staphylococcus aureus* at 2.5 mg/L DHA [14]. We diluted DHA with tryptic soy broth 1% Glucose (TSB + 1% Glu;TSB, Merck KGaA, Darmstadt, Germany). As DHA was delivered dissolved in ethanol (EtOH), we needed to use equal amounts of EtOH mixed with media as controls for biofilm formation.

### 2.2. Microorganisms and Primers

We used *Staphylococcus aureus* ATCC 29213 and one clinical isolated methicillin-resistant *Staphylococcus aureus* strain (MRSA) for biofilm formation. The MRSA strain was isolated from a patient undergoing prosthetic joint infection treatment at the Department of Orthopedics and Traumatology of the Medical University Innsbruck, Austria. This study used a protocol that was evaluated and approved by the Human Ethics Committee of the Medical University Innsbruck (AN2017-0072 371/4.24 396/5.11-4361A).

In this study, we investigated genes that are involved in biofilm formation. Table 1 shows the primers used in this study (Metabion GmbH, Planegg/Steinkirchen, Germany).

### 2.3. Biofilm Formation

The biofilms were grown on grooved titanium grade 4 cylinders with 6 mm diameter and 6 mm height (TU Braunschweig, Germany). Before use, the titanium samples were washed in sterile distilled water and sterilized with UV-light for 1 h at 254 nm with 40 μW/cm^2^ intensity (ESCO SC2-6E1, UV-30A Lamp). For each pre-culture of *Staphylococcus aureus* ATCC 29213 and the MRSA strain, 3 colonies were suspended in 2 mL of TSB + 1% Glu media in a 15 mL centrifuge tube (VWR International, Radnor, PA, USA). The pre-cultures were incubated in a moisture chamber at 37 °C on a shaker with 200 rpm for 24 h. For biofilm growth on the upper circular area of the titanium cylinders, the upper sample area was sealed off with a silicone O-ring to fit into a well of a 48-well plate. Six titanium samples were used for each strain. After incubation, the pre-cultures were diluted 1:100 with DHA-TSB + 1% Glu media and EtOH-TSB + 1% Glu for controls. A total of 500 μL of each culture dilution were pipetted in each well used. The cultures were incubated at 37 °C in a moist chamber on a shaker with 200 rpm for 24 h.

### 2.4. RNA Extraction

After 24 h of incubation for the formation of biofilms, the samples were washed in PBS, and 6 samples of each strain were put into a 15 mL centrifuge tube (VWR International, Radnor, PA, USA) and 1 mL of TRI-Reagent (TRI Reagent^®^, Sigma-Aldrich, St. Louis, MO, USA) was added. The tubes were vortexed for 15 s and sonicated for 3 min in an ultrasound bath (ultrasonic peak power: 800 W; Bactosonic, Bandelin electronic GmbH & Co. KG, Berlin, Germany). The liquids in the tubes were added to screw cap micro tubes (Screw cap micro tubes, Sarstedt AG and Co., Nümbrecht, Germany) filled with 25–50 mg of glass beads (acid-washed glass globules, Ø 0.1 mm, Carl-Roth GmbH + Co. KG, Karlsruhe, Germany). In a FastPrep-24TM 5G (MP Biomedicals, Thermo Fisher Scientific, Waltham, MA, USA), the tubes were placed and processed 3 times for 35 s at 10 m/s. The tubes were cooled on ice for 2 min in between the repetitions. Afterward, 200-μL chloroform was added to each tube, and the tubes were vortexed for 15 s and incubated at room temperature for 7 min. Afterward, the tubes were centrifuged at 4 °C and 12,000 G for 15 min. Following the centrifugation, the upper phase containing the RNA was transferred to a 1.5 mL microcentrifuge tube, and we added an equal volume of 70% cooled ethanol. The manufacturer’s procedures were followed as described in the QIAGEN Supplementary Protocol. For purification of total RNA from bacteria, RNeasy^®^ Mini Kit was used. The procedure was adapted to our demands. The processed RNA liquid for each strain was pooled on one membrane. The elution was performed with 50 μL of RNase-free water. For RNA quantification, 1 μL of the eluate was analyzed with a spectrophotometer (DeNovix DS-11 FX +μVolume Spectrophoto-/Fluorometer, Biozym Scientific GmbH, Hessisch Oldendorf, Germany). A total of 10 μg of extracted RNA were mixed together with RNase-free water to achieve a total volume of 25 μL. Then, 1 μL of DNase I (2 units) with 2.6 μL of 10xBuffer from the kit were added. The first incubation was done at 37 °C for 30 min. Afterward, 5 mM of EDTA was added to inactivate the reaction and incubated at 75 °C for 10 min. After the DNase treatment, we measured 1 μL of RNA with a spectrophotometer.

### 2.5. cDNA Synthesis from Bacterial RNA

After RNA extraction, 1 μg of RNA was used for cDNA synthesis. We used the suggested procedure by the manufacturer (iScriptTM RT Supermix, Bio-Rad Laboratories, Feldkirchen, Germany). For performing the reaction protocol, we used the PikoReal 96 System (PikoReal 96 Realtime PCR System, Thermo Fisher Scientific, Waltham, MA, USA).

### 2.6. Investigation of Influence by DHA on Bacterial Biofilm Gene Expression Using Real-Time Quantitative Polymerase Chain Reaction (Real-Time qPCR)

After the cDNA synthesis, we followed the protocol of the iQTM SYBR Green Supermix kit (iQTM SYBR Green Supermix, Bio-Rad Laboratories, Feldkirchen, Germany) with a total volume of 20 μL per reaction. A total of 10 µM stock solutions of the primers used were prepared previously (Table 1). For each real-time qPCR run, we used 0.5 μL of each forward and reverse primer stock solution. Then we added 50 ng of cDNA. The same temperature and maximum times were used as in the manufacturer’s procedures for thermal cycling (iQTM SYBR Green Supermix, Bio-Rad Laboratories, Feldkirchen, Germany). Default instrument settings were used for melting curve analysis (PikoReal 96 Realtime PCR System, Thermo Fisher Scientific, Waltham, MA, USA).

### 2.7. Statistical Analysis

For statistical analysis, we used GraphPad Prism 9. We calculated gene expression fold change with delta-delta ct values for each replicate as well as standard deviation after we calculated the average for each gene and condition. The results were then displayed in bar graphs with implemented deviations.

## 3. Results

### Influence of DHA on Expression of Biofilm-Related Genes

The ATCC 29213 strain showed at 0.625 mg/L DHA 0.3–0.7 lower gene expression levels on ica genes than the control (Figure 1A). SigB was similarly expressed as the control. Gene regulators SarA, agrA and agrC did not show significant expression levels. At a concentration of 1.25 mg/L DHA, icaADBC genes showed similar expression levels as the controls (Figure 1B). Sigma factor B showed 2.46 higher gene expression levels than its control. Gene regulator SarA showed 4.19 higher levels, while agrA showed 3.36 and agrC 4.57 higher levels of gene expression.

At 0.625 mg/L DHA concentration, MRSA showed 0.4–0.6 lower gene expression levels in ica genes than the controls (Figure 1C). SigB and gene regulators like SarA and agrA/agrC showed the same levels of gene expressions as the controls. All ica genes were less expressed in comparison with the controls at 1.25 mg/L DHA (Figure 1D). SigB showed 2.12 higher expression levels than its control, while agrA and agrC were 1.45 times and 1.19 times more expressed than their controls. We can see that sub-MIC concentrations of DHA do not increase the expression of ica genes and therefore do not enhance the production of PIA and biofilm formation.

## 4. Discussion

In this study, we evaluated the expression of ica locus genes and their regulatory genes in MRSA and ATCC 29213 after treatment with different sub-MICs of DHA.

MRSA is responsible for a 30% rate of death when occurring in MRSA bacteremia and is a great concern in modern medicine [16]. MRSA is equipped with the mecA gene that helps bacteria to withstand inhibition of cell-wall biosynthesis by beta-Lactam antimicrobial drugs. Due to its high mobile locus, the staphylococcal cassette chromosome *mec* (SCCmec), mecA can be transferred from one *Staphylococcal* strain to another via horizontal gene transfer [17]. The resistance of MRSA originates in a reduced affinity to beta-lactams by penicillin-binding Proteins (PBPs), which are essential for cell wall synthesis, and are the targeted protein of beta-lactams and their inhibitory effect against *Staphylococcus aureus* [18].

PIA plays an important role in biofilm formation and is an important pathogenic factor [19]. The analysis of the expression of PIA genes and its regulatory factors are important to understanding biofilm formation and the inhibition mechanism behind DHA on *Staphylococcus aureus*. This study showed that cell stress is present in ATCC 29213 when 1.25 mg/L DHAwas used due to increased expression of SigB. Additionally, higher gene expression occurs in SarA and agr, but they do not affect icaADBC gene expression. In MRSA also, SigB, SarA, and agr are highly expressed, but icaADBC expression is even lower than in its control at 1.25 mg/L DHA.

PIA expression is regulated by icaADBC. PIA is an extracellular poly-beta(1-6)N-acetylglucosamine. *IcaA* is an *N*-acetylglucosaminyltransferase responsible for the synthesis of PIA oligomers from UDP-N-acetylglucosamine [20]. IcaD supports IcaA and gives IcaA optimal efficiency during PIA synthesis. For externalization of PIA to the bacterial cell surface, icaC translocates the poly-N-acetylglucosamine polymer. During the translocation, icaB deacetylates the molecule [21]. It must be mentioned that PIA and their responsible genes icaADBC are not the only proteins that can lead to biofilm formation. Alternative extracellular biofilm formation proteins in *Staphylococcus aureus* are Fibronectin binding Proteins A and B [22], Protein A [23], Bap [24], and SasG [25]. These are pathogenic factors that can concur during different phases of the biofilm formation independently of the ica locus [26]. These factors are regulated by the gene SarA. Therefore, a lower gene expression in ica genes does not mean that biofilm formation is reduced, while SarA expression is increased. The inhibition mechanism of DHA is still unknown, but fatty acids have a number of inhibitory effects on bacterial cells, such as interference with oxidative phosphorylation, disruption of electron transport chain, formation of peroxidation products, enzyme inhibition, cell lysis, induction of autolysis, inhibition of nutrient uptake, leakage of cell metabolites, and inhibition of fatty acid biosynthesis [27].

This study has shown new insights into the influence of DHA on biofilm gene expression. Under the influence of DHA, biofilm formation in *Staphylococcus aureus* does not occur via the ica locus. DHA increases the stress response in both the laboratory strain ATCC 29213 and the isolated MRSA strain. This shows, in combination with previous studies about the influence of DHA on biofilm formation, that DHA does not increase biofilm formation mechanisms in sub MICs. This suggests that DHA can be used as an anti-biofilm substance and antibiotic booster substance as therapy against prosthetic joint infections. Further studies must be performed to study the use of omega-3 fatty acids as anti-biofilm substances in combination with antibiotics.

One limitation of this study is the need to use ethanol-based DHA agents. Ethanol has an influence on the expression of icaADBC genes; therefore, icaADBC gene expression is biased due to the use of ethanol in the controls of this study [28]. This study concentrated only on gene regulators of PIA expression. To consider DHA as a safe antibacterial agent, alternative biofilm proteins could be examined. As this study also uses MRSA, the influence of DHA on the expression of mecA could have been examined.

## 5. Conclusions

DHA does not enhance PIA-dependent biofilm formation but still reduces bacterial CFU in biofilms. DHA can be considered an option as a therapeutic substance against biofilm formation and may be a good tool for reducing the risk of MRSA formation. DHA can be considered as a booster substance in prosthetic joint infection patients to enhance antibiotic efficiency.

## Figures and Tables

**Figure 1 antibiotics-11-00932-f001:**
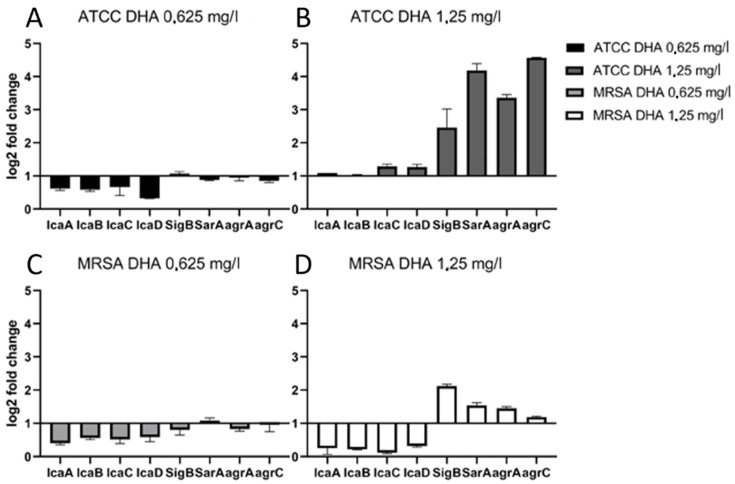
Results of icaADBC, SigB, SarA, and agrAC gene expression for biofilm produced at 0.625 mg/L DHA and 1.25 mg/L DHA by *Staphylococcus aureus* ATCC 29213 (**A**,**B**) and a clinical MRSA isolate (**C**,**D**). The results for the comparison between different DHA concentrations and their controls were analyzed by calculation of delta-delta ct values with each standard deviation.

**Table 1 antibiotics-11-00932-t001:** Forward and reverse primers for *Staphylococcus aureus* ATCC 29213 and an isolated clinical MRSA strain.

Gene	Forward Primer	Reverse Primer
ica A	5′-CGC ACT CAA TCA AGG CAT TA-3′	5′-CCA GCA AGT GTC TGA CTT CG-3′
ica B	5′-CAC ATA CCC ACG ATT TGC AT-3′	5′-TCG GAG TGA CTG CTT TTT CC-3′
ica C	5′-CTT GGG TAT TTG CAC GCA TT-3′	5′-GCA ATA TCA TGC CGA CAC CT-3′
ica D	5′-ACC CAA CGC TAA AAT CAT CG-3′	5′-GCG AAA ATG CCC ATA GTT TC-3′
SigB	5′-AGA AGC AAT GGA AAT GGG AC-3′	5′-CTT AAA CCG ATA CGC TCA CC-3′
SarA	5′-AAG GAC AAT CAC ATC ACG AAG-3′	5′-GAA CGC TCT AAT TCA GCG G-3′
agrA	5′-GCC CTC GCA ACT GAT AAT CC-3′	5′-CAT CGC TGC AAC TTT GTA GAC-3′
agrC	5′-GCA TCA ACT GAA ATT GAT GAC C-3′	5′-ACC TAA ACC ACG ACC TTC AC-3′
16s	5′-GAA AGC CAC GGC TAA CTA CG-3′	5′-CAT TTC ACC GCT ACA CAT GG-3′

## Data Availability

All data are presented in the publication.

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
