# Peer review of "Antibiofilm Activity of Omega-3 Fatty Acids and Its Influence on the Expression of Biofilm Formation Genes on Staphylococcus aureus"

_antibiotics, 2022, doi:10.3390/antibiotics11070932_

Round 1
Reviewer 1 Report
Manuscript #antibiotics-1795821 entiteled "Antibiofilm activity of omega-3 fatty acids and its influence on the expression of biofilm formation genes on Staphylococcus aureus" by Spiegel et al presents (in line with title) the effect of DHA on the expression of several genes in biofilm S. aureus. In my opinion this rather short study should be presented as Short Communication, not an Article. The results and general idea are quite interesting though. Below, some issues are mentioned:
L12-3: "Staphylococcus aureus" or "Staphylococcus epidermis" should be italicized. Please check throught the manuscript.
L43: should "coagulase negative", "CNS" and "MRSA" be italicized? it is not latin. Please check throught the manuscript. Similarly "ATCC 29213" in L96.
Table 1 has no caption. Similarly, Figure 1.
The authors stated that "One limitation of this study is the need to use ethanol-based DHA agents. Ethanol has an influence on the expression of icaADBC genes therefore icaADBC gene expression is biased due to the use of ethanol in the controls of this study (28)." Was the blank performed herein? The effect of same EtOH concentrations alone without the xenobiotic ?
Author Response
RESPONSE TO THE REVIEWERS
Dear Editors
Guest Editor: Journal – Antibiotics
Please find enclosed a carefully revised version of our manuscript entitled, “ANTIBIOFILM ACTIVITY OF OMEGA-3 FATTY ACIDS AND ITS INFLUENCE ON THE EXPRESSION OF BIOFILM FORMATION GENES ON STAPHYLOCOCCUS AUREUS” - antibiotics-1795821” for re-submission to “Antibiotics – Special Issue: Strategies to Boost Antibiotic Activity”.
We were very pleased with the positive evaluation of our manuscript by the editors and the reviewers. However, the added some comments and suggestions, which were all addressed in the revised version of our paper as detailed in the point-to-point response.
We hope that our paper is now acceptable for publication in your journal.
Yours sincerely,
Christopher Spiegel
Corresponding Author
Reviewer 1
Manuscript #antibiotics-1795821 entiteled "Antibiofilm activity of omega-3 fatty acids and its influence on the expression of biofilm formation genes on Staphylococcus aureus" by Spiegel et al presents (in line with title) the effect of DHA on the expression of several genes in biofilm S. aureus. In my opinion this rather short study should be presented as Short Communication, not an Article. The results and general idea are quite interesting though. Below, some issues are mentioned:
Thank you for the comments.
Point 1:L12-3: "Staphylococcus aureus" or "Staphylococcus epidermis" should be italicized. Please check throught the manuscript.
Response 1: Thank you for your feedback. The manuscript has been checked and modified.
Point 2: L43: should "coagulase negative", "CNS" and "MRSA" be italicized? it is not latin. Please check throught the manuscript. Similarly "ATCC 29213" in L96.
Response 2: Thank you for your feedback. We applied your suggestions to the manuscript.
Point 3:Table 1 has no caption. Similarly, Figure 1.
Response 3: The mentioned missing captions have been included into the script.
Point 4:The authors stated that "One limitation of this study is the need to use ethanol-based DHA agents. Ethanol has an influence on the expression of icaADBC genes therefore icaADBC gene expression is biased due to the use of ethanol in the controls of this study (28)." Was the blank performed herein? The effect of same EtOH concentrations alone without the xenobiotic ?
Response 4: The controls in this study have been performed with the equal amount of EtOH as the DHA containing samples. 1.25 mg/L DHA samples had a concentration of 5mg/L of EtOH.

Reviewer 2 Report
We suggest some major and minor revision to improve the readness of the paper:
- improve the introduction, this part is well written but it is not clear from the text why the effect of DHA has an anti-biofilm formation effect in the MRSA strain and not in the ATCC29213 strain.
- improve the results,
- clarified the biological importance of the study,
Minor:
- To write names of the strains in italics
- To write the associated acronyms and the corresponding names and always use the same wording along the text
- In materials and methods, it would be advisable to add details in the table showing the genes. Additional information regarding the annealing temperatures of the primers and other details on the PCR reaction could be included.
- Line 165-166 shows “250ng of each forward and reverse primer”. The known concentration of the primers is that of the stocks, expressed in mM. The unit of measurement reported should be adjusted. Furthermore, the concentrations of both primers and cDNA are very high for a sensitive system such as real time PCR, qRT-PCR.
- A figure could be inserted that encompasses the entire workflow
- A structural figure of the molecule tested in the study could be included, docosaexanoic acid DHA
- -In the results, figure 1 must be arranged by inserting the letters (a, b, c, d). By doing so, the figure can be easily commented along the text. Rewrite the caption accordingly.
- Starting from line 217, some data are commented that are not reported along the text. Report the data in such a way as to make the concept clearer
- To write why the administration of DHA only affects the MRSA strain and not the ATCC 29213 strain
- To explain better the hypothesis that the lower concentration (0.625 mg / ml) is more effective than the higher concentration tested.
- Since the limits in the effective use of DHA are reported throughout the text, propose valid alternatives for the effective use of the molecule. For example, intermediate dilutions could be made to lower the ethanol concentration and the toxicity of the molecule
Author Response
RESPONSE TO THE REVIEWERS
Dear Editors
Guest Editor: Journal – Antibiotics
Please find enclosed a carefully revised version of our manuscript entitled, “ANTIBIOFILM ACTIVITY OF OMEGA-3 FATTY ACIDS AND ITS INFLUENCE ON THE EXPRESSION OF BIOFILM FORMATION GENES ON STAPHYLOCOCCUS AUREUS” - antibiotics-1795821” for re-submission to “Antibiotics – Special Issue: Strategies to Boost Antibiotic Activity”.
We were very pleased with the positive evaluation of our manuscript by the editors and the reviewers. However, the added some comments and suggestions, which were all addressed in the revised version of our paper as detailed in the point-to-point response.
We hope that our paper is now acceptable for publication in your journal.
Yours sincerely,
Christopher Spiegel
Corresponding Author
Reviewer 2
Point1: - improve the introduction, this part is well written but it is not clear from the text why the effect of DHA has an anti-biofilm formation effect in the MRSA strain and not in the ATCC29213 strain.
Response 1: Thank you for your feedback. Just to clarify, the information about the different activity of DHA on MRSA and on ATCC29213 strains is not stated in the introduction. In the discussion section, we stated the following: “This study has shown new insights on the influence of DHA on biofilm gene expression. Under the influence of DHA, biofilm formation in Staphylococcus aureus does not occur via the ica locus. DHA is increasing the stress response in both the laboratory strain ATCC 29213 and the isolated MRSA strain. This shows, in combination with previous studies about the influence of DHA on biofilm formation, that DHA is not increasing biofilm formation mechanisms in sub MICs. This suggests that DHA can be used as anti-biofilm substance and antibiotic booster substance as therapy against prosthetic joint infections”.
Point 2: - improve the results,
Response 2: It was not clear by the reviewer 2 what exactly might be done to improve the results section in our manuscript. In this case, we checked the whole Results and carried out some corrections and improvement, as we considered necessary.
Point 3: - clarified the biological importance of the study,
Response 3: Thank you for the comment. The biological as well as clinical importance of the study was already stated in the manuscript: “This study has shown new insights on the influence of DHA on biofilm gene expres-sion. Under the influence of DHA, biofilm formation in Staphylococcus aureus does not oc-cur via the ica locus. DHA is increasing the stress response in both the laboratory strain ATCC 29213 and the isolated MRSA strain. This shows, in combination with previous studies about the influence of DHA on biofilm formation, that DHA is not increasing bio-film formation mechanisms in sub MICs. This suggests that DHA can be used as an-ti-biofilm substance and antibiotic booster substance as therapy against prosthetic joint infections. Further studies must be performed to study the use of omega-3 fatty acids as anti-biofilm substance in combination with antibiotics.”
Point 4: To write names of the strains in italics
Response 4: Thank you for your feedback. The manuscript has been checked and modified.
Point 5: To write the associated acronyms and the corresponding names and always used the same wording along the text
Response 5: Thank you for your feedback. We applied your suggestions to the manuscript.
Point 6: In materials and methods, it would be advisable to add details in the table showing the genes. Additional information regarding the annealing temperatures of the primers and other details on the PCR reaction could be included.
Response 6: Thank you for the suggestion. We improved the Materials and Methods and included additional information, as we considered necessary and useful for the audience.
Point 7: Line 165-166 shows “250ng of each forward and reverse primer”. The known concentration of the primers is that of the stocks, expressed in mM. The unit of measurement reported should be adjusted. Furthermore, the concentrations of both primers and cDNA are very high for a sensitive system such as real time PCR, qRT-PCR.
Response 7: Thank you for your constructive suggestion. We added mM concentrations of stock solutions and the amount of primers used in this study.
Point 8: A figure could be inserted that encompasses the entire workflow
Response 8: Thank you for the suggestions. As we already detailed each step on the Materials and Methods section, we decided not to insert a workflow figure otherwise; it could be repetitive for the readers.
Point 9: A structural figure of the molecule tested in the study could be included, docosaexanoic acid DHA
Response 9: Thank you for the suggestion.
Point 10: In the results, figure 1 must be arranged by inserting the letters (a, b, c, d). By doing so, the figure can be easily commented along the text. Rewrite the caption accordingly.
Response 10: Thank you a lot for this advice. We added letters and rewrote the caption accordingly.
Point 11: Starting from line 217, some data are commented that are not reported along the text. Report the data in such a way as to make the concept clearer
Response 11: We are grateful for the suggestion of yours. We changed the report of the data.
Point 12: To write why the administration of DHA only affects the MRSA strain and not the ATCC 29213 strain
Response 12: DHA does affect both strains but ATCC29213 shows a higher gene fold change of stress gene at 1.25 mg/L DHA than MRSA.
Point 13: To explain better the hypothesis that the lower concentration (0.625 mg / ml) is more effective than the higher concentration tested.
Response 13: 0.625 mg/L DHA may be better against biofilm formation due to no increase in biofilm formation by icaADBC but 0.625 mg/L DHA also showed lower inhibition potential in a previous study of Coraca-Huber et.al.
Point 14: Since the limits in the effective use of DHA are reported throughout the text, propose valid alternatives for the effective use of the molecule. For example, intermediate dilutions could be made to lower the ethanol concentration and the toxicity of the molecule
Response 14: EtOH concentration in 1.25 mg/L DHA has been 5 mg/L. This low amount of EtOH has no antimicrobial activity.

Round 2
Reviewer 1 Report
The manuscript has been revised in a satisfactory way.
Reviewer 2 Report
The authors clearly responded to the questions and revised the paper.